# Cardiogenic Shock Integrated PHenotyping for Event Reduction: A Pilot Metabolomics Analysis

**DOI:** 10.3390/ijms242417607

**Published:** 2023-12-18

**Authors:** Nuccia Morici, Gianfranco Frigerio, Jonica Campolo, Silvia Fustinoni, Alice Sacco, Laura Garatti, Luca Villanova, Guido Tavazzi, Navin K. Kapur, Federico Pappalardo

**Affiliations:** 1Dipartimento Cardio-Respiratorio, IRCCS Fondazione Don Carlo Gnocchi ONLUS, 20100 Milan, Italy; 2Department of Clinical Sciences and Community Health, University of Milan, Fondazione IRCCS Ca’ Granda Ospedale Maggiore Policlinico, 20126 Milan, Italy; gianfranco.frigerio@uni.lu (G.F.); silvia.fustinoni@unimi.it (S.F.); 3Luxembourg Centre for Systems Biomedicine (LCSB), University of Luxembourg, 4365 Luxembourg, Luxembourg; 4Institute of Clinical Physiology CNR, 20162 Milan, Italy; 5Intensive Cardiac Care Unit and De Gasperis Cardio Center, ASST Grande Ospedale Metropolitano Niguarda, 20162 Milan, Italy; 6Unit of Anaesthesia and Intensive Care, Department of Clinical-Surgical, Diagnostic and Paediatric Sciences, University of Pavia, 27100 Pavia, Italy; 7Anesthesia and Intensive Care, Fondazione Policlinico San Matteo Hospital IRCCS, Anestesia e Rianimazione I, 27100 Pavia, Italy; 8The CardioVascular Center, Tufts Medical Center, Boston, MA 02111, USA; nkapur@tuftsmedicalcenter.org; 9Cardiothoracic and Vascular Anesthesia and Intensive Care, AO SS. Antonio e Biagio e Cesare Arrigo, 15100 Alessandria, Italy; fedepappa@me.com

**Keywords:** cardiogenic shock, biomarkers, metabolomics, glycocalyx

## Abstract

Cardiogenic shock (CS) portends a dismal prognosis if hypoperfusion triggers uncontrolled inflammatory and metabolic derangements. We sought to investigate metabolomic profiles and temporal changes in IL6, Ang-2, and markers of glycocalyx perturbation from admission to discharge in eighteen patients with heart failure complicated by CS (HF-CS). Biological samples were collected from 18 consecutive HF-CS patients at admission (T0), 48 h after admission (T1), and at discharge (T2). ELISA analytical techniques and targeted metabolomics were performed Seven patients (44%) died at in-hospital follow-up. Among the survivors, IL-6 and kynurenine were significantly reduced at discharge compared to baseline. Conversely, the amino acids arginine, threonine, glycine, lysine, and asparagine; the biogenic amine putrescine; multiple sphingolipids; and glycerophospholipids were significantly increased. Patients with HF-CS have a metabolomic fingerprint that might allow for tailored treatment strategies for the patients’ recovery or stabilization.

## 1. Introduction

Cardiogenic shock (CS) is a life threatening, heterogenous syndrome of end-organ hypoperfusion that can potentially lead to multiorgan failure if not promptly recognized and adequately treated [1,2]. CS epidemiology has been exclusively focused on patients with acute myocardial infarction (AMI), although recent data have shown that CS related to AMI and to heart failure (HF) is approximately equally distributed [1]. The use of translational medicine, including immunophenotyping, metabolomics, inflammatory biomarkers, and genomics, is emerging as a robust approach to tailor HF diagnosis and treatment, yet it has been scarcely applied in CS.

The aim of this project was to explore the biomarkers and metabolomic profile of a cohort of patients with HF-CS. The Cardiogenic shock Integrated PHenotyping for Event Reduction (CIPHER) study (ClinicalTrials.gov Identifier: NCT04323371) is a prospective pilot study that is part of the Italian Altshock-2 program. The inclusion and exclusion criteria have been previously reported [3]. This study was approved by the Local Ethics Committee of Milano Area 3 of the ASST Grande Ospedale Metropolitano Niguarda (Piazza Ospedale Maggiore 3, 20162 Milano). All patients provided written informed consent. This study’s endpoints were the following: (i) the exploratory assessment of targeted metabolomics through the quantification of almost 180 molecules, including acylcarnitine, amino acids and biogenic amines, hexoxide, sphingolipids, and glycerophospholipids; (ii) the exploratory assessment of temporal changes in IL6, Ang-2, and markers of glycocalyx perturbation from admission to discharge.

## 2. Results

The median age of the study population was 50 years (25% and 75% percentiles equal to 42 and 60 years, respectively), and the study population was predominantly males (78%) (Table 1). We assessed four different parameters involved in the pro-inflammatory response (interleukin-6 (IL-6)) and in the endothelial perturbation (angiopoietin-2, syndecan-1, and heparan sulfate) (Appendix A). Among these biomarkers, the pro-inflammatory cytokine IL-6 was significantly reduced compared to baseline in the patients who were successfully hemodynamically stabilized with medical treatment (35.24 ± 14.79 pg/mL versus 52.43 ± 26.09 pg/mL, *p* = 0.016). No significant differences were reported at baseline among the survivors and non-survivors (during hospital stay).

The results of the mixed linear models for the metabolomic analysis are reported in the Appendix A. Although no significant differences in the metabolomic profile were found when comparing T1 vs. T0, several metabolites were significantly different comparing T2 vs. T0 (Figure 1). Kynurenine, N-acetylornithine, and glutamic acid were the metabolites for which significant lower levels were found in T2 compared to T0, while several metabolites were significantly higher in T2 vs. T0, among them the amino acids arginine (Arg), threonine (Thr), glycine (Gly), lysine (Lys), and asparagine (Asn); the biogenic amine putrescine; 10 sphingolipids; and a total of 69 glycerophospholipids (among which were 8 lyso PC, 28 PC aa, and 33 PC ae). The results of the pathway analyses showed that, among the potential most-altered pathways, are glycine, serine, and threonine metabolism; arginine and proline metabolism; and linoleic acid metabolism (Appendix A). Furthermore, when considering only the samples collected at T0, a few glycerophospholipids (PC aa C40:2, PC aa C34:1, PC ae C44:6, PC ae C38:1, PC aa C32:2) were lower in non-survivors compared to patients who survived (odds ratio of the logistic regressions < 1), while putrescine was higher (odds ratio > 1), although not statistically significant when considering the FDR *p*-value (Appendix A).

## 3. Methods and Experimental Design

Blood samples were collected from 18 consecutive HF-CS patients admitted to the Intensive Coronary Care Unit (ICCU) at ASST Grande Ospedale Metropolitano Niguarda, Milan, and locally stored until all needed samples were obtained.

Three blood samples were collected from each patient, at admission (T0), 48 h after admission (T1), and at discharge (T2), after a median of 44 (interquartile range 32–58) days. The biomarkers were assessed with ELISA analytical techniques; comparisons between T0 and T1 were performed with the paired t-test or Wilcoxon signed-rank test, while comparisons among T0, T1, and T2 were performed with the non-parametric Friedman test. The metabolomic profile of plasma samples was assessed with a targeted approach, in particular, a liquid chromatography tandem mass spectrometry method implementing the AbsoluteIDQ p180 kit (Biocrates Life Sciences AG, Innsbruck, Austria) [4]. With this assay, a total of 188 metabolites were quantified: a total of 21 amino acids, 21 biogenic amines, the sum of hexose (H1), 40 acylcarnitine, 15 sphingolipids, and 90 glycerophospholipids among which were 14 lysophosphatidylcholines (LysoPC), 38 diacylphosphatidylcholine (PC aa), and 38 acylalkylphosphatidylcholine (PC ae). This approach is widely used in the metabolomic community and has the advantage of good interlaboratory reproducibility. The analytical details used in our analyses were extensively reported previously [5]. Metabolite concentrations were log-transformed and standardized, and then, a linear mixed-effects model was built for each metabolite in which the dependent variable was the metabolite concentration; the independent variables with fixed effects were age, body mass index, sex, and sample collection (T0, T1, or T2), while the patients were considered the random intercept variable. In addition, logistic regression models were built to assess the association between metabolite concentration at T0 and patient survival; for each metabolite, a model was built considering whether the patient died (yes or no) as the dependent variable, and the metabolite measured at T0 as the independent variable along with sex, age, and body mass index as further independent variables. The inclusion of all these independent variables to the model was necessary, despite the low number of observations, since the metabolome is highly influenced by several confounding factors, and we decided to correct at least for the most important ones. The *p*-values were adjusted for multiple testing controlling the false discovery rate (FDR), and an FDR *p*-value lower than 0.1 was considered statistically significant. Finally, pathway analyses were performed using MetaboAnalyst [6] with the global test enrichment method, the topology analysis out-degree centrality, and the pathway library Homo sapiens (KEGG).

## 4. Discussion

For the first time to our knowledge, we highlight the prominent role of systemic inflammation in HF-CS using a metabolomics-based approach as underlined with the increased values of the serum tryptophan–kynurenine pathway metabolites, which consistently decreased in the survivors.

If confirmed in larger studies, the evaluation of inflammatory and metabolomic profiles might turn into an innovative fingerprint to drive patient-tailored stratification in cardiogenic shock and as a tool to titrate the amount of hemodynamic support.

Cardiogenic shock (CS) is a heterogenous syndrome with in-hospital mortality of up to 50% that has remained stagnant over time despite observed improvements with pharmacological and non-pharmacological approaches [1]. This lack of benefit has been attributed to the inability to characterize the different phenotypes with specific responses to treatments. Accordingly, it has been suggested to move on from diagnostic and therapeutic strategies based on the improvement of cardiac output toward mechanistic drivers of shock phenotypes that allow for a more personalized patient selection for treatment strategies based on integrative approaches, including metabolomics and biomarkers of inflammation and endothelial dysfunction [2].

Several biomarkers have been previously investigated, but they were mostly limited to acute coronary syndrome (ACS) patients and provided limited information (Table 2).

Inflammatory mediators, including interleukin-6 (IL-6) and tumor necrosis factor-alpha (TNF-α), are frequently elevated in CS and add further to cardiac dysfunction through their negative inotropic effect. In addition, cytokines lead to the production of high levels of nitric oxide (NO) through the induction of inducible nitric oxide synthase (iNOS), which may result in a state of inappropriate vasodilation and in perturbation of endothelial function [1]. Inflammatory stimuli applied to the endothelium may, therefore, contribute to the alteration of other regulatory systems. Among several biomarkers, the endothelial glycocalyx (through its most prevalent proteoglycan syndecan-1) and the Ang/Tie system are involved in the vascular barrier dysfunction during critical illness and are associated with the development of CS in acutely ischemic patients [7,8,9,10,11,12]. However, their prognostic role is not clear across the spectrum of CS etiologies.

Moreover, the biochemical pathways involved in heart failure (HF) suggest that, as hearts begin to fail, altered energetics play an increasingly important role in pathogenesis. The concept that unique patterns of metabolomic expression and energy utilization occur in different etiologies and severity grading of HF has recently been introduced [13] and could revolutionize the diagnosis and management of heart failure.

Metabolites can be considered a direct signature of biochemical activity [14]. A few pathways have been explored in heart failure but are not defined across the whole spectrum of CS syndrome. Preliminary data exist on structural abnormalities in mitochondria along with reduced activity of the respiratory carriers (Krebs cycle intermediates) and oxidative phosphorylation [13,15].

They can mostly occur in patients with CS after longstanding disease in order to enhance the patients’ tolerance to low cardiac output states and/or elevated ventricular filling pressures. As a proof of concept, metabolomic profiling following left ventricular (LV) assist device implantation in end-stage HF patients resulted in a decrease in circulating L-C acylcarnitine [16] due to the reduced update and mitochondrial oxidative metabolism of FAs.

The tryptophan–kynurenine pathway metabolites have been previously associated with heart failure and atrial fibrillation [17,18,19], higher risk of acute coronary syndrome, and cardiovascular death [20,21].

It has also been reported that the tryptophan–kynurenine pathway has a tight interplay with the cytokines activation and has been implicated in the modulation of inflammatory responses in vascular and immune cells [22].

The main limitation of this study is its small sample size, which may restrict the generalizability of our results to a broader population. However, this work serves as a foundation for future studies with larger and more diverse cohorts, which can further validate and build upon our findings. Indeed, this work confirms the relevant role of metabolomic profiles in HF-CS patients. Further studies focusing on the integration with genomics and other ‘omics data and clinical phenotype will have a meaningful impact on patient outcomes.

## Figures and Tables

**Figure 1 ijms-24-17607-f001:**
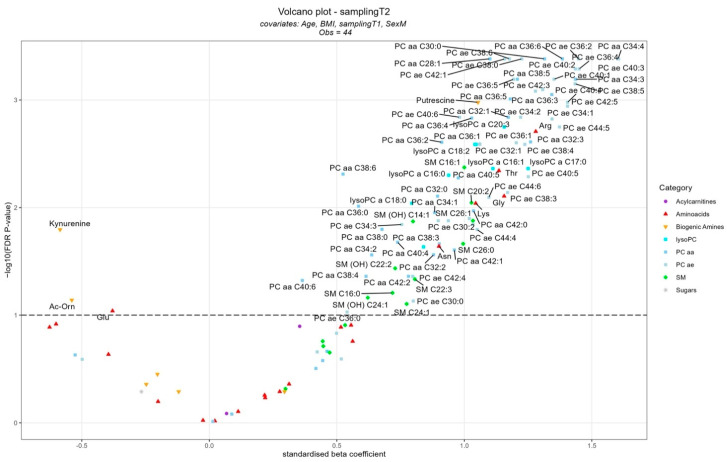
Volcano plot representing the results of the mixed effects linear regression models considering the metabolites (dependent variables) in relation to sampling (T2 vs. T0), adjusted for age, BMI, sex, and time sampling. Patients were considered as the random intercept variable. Each dot represents a metabolite, and they are displayed based on the standardized beta coefficient (x-axis) and the negative logarithm (base 10) of the FDR *p*-value (y-axis). The dashed line represents an FDR *p*-value equal to 0.1. The significantly different metabolites comparing T2 vs. T0 are noted with their full names in the graph.

**Table 1 ijms-24-17607-t001:** Baseline characteristics of the included patients (n = 18).

Variable	Number (%)
Male gender	14 (78%)
Age	50 (42–60)
BMI	25 (23–27)
Hypertension	3 (17%)
Diabetes	2 (11.1%)
Ischemic etiology	6 (33%)
On admission	
SCAI Stage	
B	6 (33%)
C	9 (50%)
D	3 (17%)
Ejection fraction (%)	20 (18–30)
Central venous pressure (mmHg)	10 (4–15)
Wedge pressure (mmHg)	19 (10–24)
Cardiac index L/min/m^2^	2 (1.4–2.5)
PAPm (mmHg)	33 (26–35)
SOFA score	5.5 (3–6)
Hb, gr/dl	9 (8–10)
Serum creatinine, mg/dl	1.7 (1.3–2.1)
ScVO2 (%)	45 (41–60)
Lactates mmol/L	3 (2.2–3.4)
During Hospital stay	
Epinephrine (17 patients)	
maximum dose, mcg/Kg/min	0.05 (0.04–0.12)
total time of administration, days	9 (5–20)
Sodium Nitroprusside (15 patients)	
maximum dose, mcg/Kg/min	0.4 (0.3–0.5)
total time of administration, days	5 (2–8)
Maximum inotropic score	7 (4–12)
IABP	13 (72%)
Total support time, days	8 (5–19)
ECMO	3 (17%)
Total support time, days	7 (6–13)
Mechanical ventilation	7 (39%)
NIV	14 (78)
CRRT	2 (11%)
In-hospital Outcome	
Heart transplantation	8 (44%)
Left ventricular assist device	1 (5%)
Death	7 (44%)

Data are presented as n (%) for categorical variables and median (25%, 75% percentiles) for continuous variables. Abbreviations. BMI: body mass index; SCAI: Society for Cardiovascular Angiography and Interventions classification; PAPm: mean pulmonary arterial pressure; SOFA: sequential organ failure assessment; Hb: hemoglobin; ScvO2: central venous oxygen saturation; IABP: intraortic balloon pump; ECMO: Extracorporeal membrane oxygenation; NIV: Non-invasive ventilation; CRRT: continuous renal replacement therapy.

**Table 2 ijms-24-17607-t002:** Biomarkers in cardiogenic shock patients.

Author	Year	Sample Size	Biomarkers	Outcome
Appoloni O Chest. 2004 Jun;125(6):2232-7	2004	33 patients (75% ischemic heart disease)	TNF- IL-6 IL-10 TGF- IFN- cytokine polymorphisms	70% ICU mortality The rare TNF-2 allele of the TNF-promoter is a strong independent factor associated with better survival from cardiogenic shock.
Geppert A Crit Care Med. 2006 Aug;34(8):2035-42	2006	38 AMI-CS patients	IL-1 IL-6 IL-10 ICAM-1 E-selectin	40% mortality at 30 days Association with IL6, no other markers. 200 pg/mL as the most valuable IL-6 cut-off concentration for predicting 30-day mortality with a specificity of 87% and a sensitivity of 74%.
Debrunner M Clin Res Cardiol. 2008 May;97(5):298-305	2008	41 AMI patients (19 with CS: 7 developed SIRS)	TNF-α IL-6 IL-1 Ra	71% In-hospital mortality in group 3 (SIRS) IL-1Ra showed the most impressive changes.
Sleeper LA Am Heart J. 2010 Sep;160(3):443-50	2010	1217 AMI patients (294 from the randomized trial and 923 from the registry)	Creatinine ≥ 1.9 mg/dL	57% in-hospital mortality at 30 days
Prondzinsky R Clin Res Cardiol. 2012 May;101(5):375-84	2012	40 AMI-CS patients	IL-1b IL-6 IL-7 IL-8 IL-10	32% mortality at 96 h The pro- and anti-inflammatory markers IL-6, IL-7, IL-8, and IL-10 showed a predictive power for mortality of infarct-related CS patients, while IL-1b did not discriminate. No IABP effect.
Link A Eur Heart J. 2013 Jun;34(22):1651-62	2013	96 CS patients (58% AMI)	Ang-1 Ang-2	37.5% mortality at 28 days. 61.5% mortality at 1 year Ang-2 level >2500 pg/mL at admission is an independent predictor for 1-year mortality (HR 2.11; 95% CI (1.03–4.36); *p* = 0.042)
Fuernau G Crit Care. 2014 Dec 21;18(6):713	2014	600 patients in the original trial (190 included: Leipzig cohort with blood sample available) AMI	OPG GDF-15	40.2% mortality at 30 days Multivariate: GDF-15, TIMI flow grade < 3 after PCI, age, LVEF, and serum lactate remained significant predictors of 30-day mortality.
Poss J Eur J Heart Fail. 2015 Nov;17(11):1152-60	2015	600 patients in the original trial (1890 included) AMI	Ang-2 IQR	41% mortality at 30 days 53% mortality at 1 year Ang-2 was an independent predictor of 30-day and 1-year mortality (with creatinine, lactate, NTproBNP, FGF 23, SAPSII, age, EF)
Fuernau G Int J Cardiol. 2015 Jul 15; 191:159-66	2015	600 patients in the original trial (190 included: Leipzig cohort with blood sample available) AMI	Creatinine NGAL KIM1 CysC	54% mortality at 1 year Creatinine demonstrated a better predictive performance at all 3 time points with respect to 1-year mortality in comparison to the other 3 biomarkers.
Poss J J Am Coll Cardiol. 2017 Apr 18;69(15):1913-1920	2017	600 patients in the original trial (480 included) AMI Validation in IABP shock II registry (188 pts) and Cardshock cohort (219 pts)	Glucose Creatinine Arterial Blood Lactate	41% mortality at 30 days Age > 73 years, prior stroke, glucose at admission > 10.6 mmol/L (191 mg/dl), creatinine at admission > 132.6 mmol/l (1.5 mg/dl), thrombolysis in myocardial infarction flow grade < 3 after percutaneous coronary intervention, and arterial blood lactate at admission >5 mmol/l.
Tolppanen H Crit Care Med. 2017 Jul;45(7):e666-e673	2017	145 ACS-CS patients	sST2 NT-proBNP	43% mortality at 30 days Combination of results for soluble ST2 and NT-proBNP provide early risk assessment beyond clinical variables of Cardshock score
Rueda F Eur Heart J. 2019 Aug 21;40(32):2684-2694	2019	48 AMI patients (Validation in 97 patients from Cardshock–71% ACS)	Quantitative proteomics analysis (QPC)	37.1% mortality at 90 days 4-protein combination: the CS4P with proteins liver-type fatty acid-binding protein (L-FABP, P07148), beta-2-microglobulin (B2MG, P61769), fructose-bisphosphate aldolase B (ALDOB, P05062), and SerpinG1 as the best protein classifier to identify short-term mortality risk with an AUC of 0.83 (95% CI 0.74–0.89)
Peng Y Clin Chim Acta. 2020 Dec; 511:97-103	2020	707 of critically ill patients with CS (55% CAD)	Systemic immune-inflammatory index (SII)	40% mortality at 30 days 50% mortality at 90 days 60% mortality at 1 year High-SII group independently associated with mortality (any time). Low-SII group (<82.85), 235 in the mid-SII group (82.8–111.7), and 236 in the high-SII group (>111.7)
Cuinet J Sci Rep. 2020 May 6;10(1):7639	2020	24 patients (12 ACS-CS—50% 6 HF-CS—25%)	WBC counts IL-1β IL-5 IL-6 IL-10 TNFα IFNγ MCP-1 Eotaxin (CCL11)	21% In-hospital mortality Early (T1) neutrophilia and IL-6, IL-10, and MCP-1, rise of eosinophils over time. Most severe shock had reduced lymphocytes and monocytes at T2 and T3
Ceglarek U Eur Heart J. 2021 Jun 21;42(24):2344-2352	2021	458 patients (derivation cohort of culprit shock, 152 patients among 458 were used for internal validation + 163 patients (validation cohort) IABP shock II trial) AMI	Lactate Cystatin C NT-proBNP IL-6	43.4% mortality at 30 days
Zhang Z Int J Gen Med. 2021 Aug 12; 14:4459-4468	2021	1487 patients (CAD 63.8%)	Lymphocyte to monocyte ratio (LMR)	Approximately 50% in-hospital mortality at 30 days Low-LMR group had a poor prognosis in crude cohort (HR: 1.40, 95% CI: 1.12–1.74, *p* = 0.003). After PSM, low-LMR group had a similar poor prognosis in crude cohort (HR: 1.31, 95% CI: 1.08–1.68, *p* = 0.016) compared with high-LMR group.

Acronyms: TNF = tumor necrosis factor; IL = interleukin; IL-1 Ra = interleukin 1 receptor antagonist; TGF = transforming growth factor; IFN = interferon; ICAM = soluble intercellular adhesion molecule; Ang = angiopoietin; OPG = osteoprotegerin; GDF = growth-differentiation factor; NGAL = neutrophil gelatinase-associated lipocalin; KIM1 = kidney injury molecule 1; CysC = cystatin C; WBC = white blood cells.

## Data Availability

Data will be available upon specific request to the corresponding author.

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
