# Peer review of "Cardiogenic Shock Integrated PHenotyping for Event Reduction: A Pilot Metabolomics Analysis"

_ijms, 2023, doi:10.3390/ijms242417607_

Round 1

Reviewer 1 Report

Comments and Suggestions for Authors

In this manuscript, authors tried to explore the biomarkers and metabolomics profile of a cohort of patients with HF-CS. However, the quality of presentation is low and not enough to support their conclusion. The manuscript can't be accepted without a major revision. I will lay some points below.

1. Authors should provide all data about the concentration of IL-6, angiopoietin-2, syndecan-1 and Heparan Sulfate at three time points.

2. The metabolites which were significantly different comparing T2 vs T0 should be highlighted in Figure 1 of the revised manuscript.

3. Please show the results of the pathway analyses with a figure or table.

Comments on the Quality of English Language

There are some spelling mistakes in the current version of manuscript. For instance, in the abstract, the word "xperimental" should be "experimental".

Author Response

Reviewer #1:

  1. Authors should provide all data about the concentration of IL-6, angiopoietin-2, syndecan-1 and Heparan Sulfate at three time points.

Reply: Thanks. A supplementary Table 1 has been added with all the concentrations at three time points.

  1. The metabolites which were significantly different comparing T2 vs T0 should be highlighted in Figure 1 of the revised manuscript.

Reply: The metabolites significantly different are highlighted as all the dots above the dashed line represent a significant metabolite, and the name of those is displayed in the figure. To further improve clarity, we added this statement to the figure legend: “The significantly different metabolites comparing T2 vsT0 are noted with their full names in the graph.”

  1. Please show the results of the pathway analyses with a figure or table.

Reply: Thanks for the suggestion, the complete results of the pathway analyses have now been reported in the Supplementary Table 3.

Reviewer 2 Report

Comments and Suggestions for Authors

Cardiogenic shock Integrated PHenotyping for Event Reduc- 2 tion: a pilot metabolomics analysis

Cardiogenic shock (CS) is a life threatening, heterogenous syndrome of end-organ hypoperfusion potentially leading to multiorgan failure if not promptly recognized and adequately treated. References are relevated to the research. The design research and the results including tables and figures are adequately described. The main limitation of this study is ithe small sample size, which may restrict the generalizability of results to a broader population. Conclusions are supported by the results.

Author Response

Reviewer #2:

Cardiogenic shock (CS) is a life threatening, heterogenous syndrome of end-organ hypoperfusion potentially leading to multiorgan failure if not promptly recognized and adequately treated. References are relevated to the research. The design research and the results including tables and figures are adequately described. The main limitation of this study is ithe small sample size, which may restrict the generalizability of results to a broader population. Conclusions are supported by the results. There are some spelling mistakes in the current version of manuscript. For instance, in the abstract, the word "xperimental" should be "experimental".

Reply: Thanks. The spelling mistakes have been removed.